# Immune-Related Genomic Schizophrenic Subtyping Identified in DLPFC Transcriptome

**DOI:** 10.3390/genes13071200

**Published:** 2022-07-04

**Authors:** Eva Childers, Elijah F. W. Bowen, C. Harker Rhodes, Richard Granger

**Affiliations:** 1Dartmouth College, Hanover, NH 03755, USA; eva.p.childers.gr@dartmouth.edu (E.C.); elijah.floyd.william.bowen@dartmouth.edu (E.F.W.B.); 2NeuroDex, Inc., Natick, MA 01760, USA; charkerrhodes@gmail.com

**Keywords:** inflammation, schizophrenia, transcriptome, subtypes

## Abstract

Well-documented evidence of the physiologic, genetic, and behavioral heterogeneity of schizophrenia suggests that diagnostic subtyping may clarify the underlying pathobiology of the disorder. Recent studies have demonstrated that increased inflammation may be a prominent feature of a subset of schizophrenics. However, these findings are inconsistent, possibly due to evaluating schizophrenics as a single group. In this study, we segregated schizophrenic patients into two groups (“Type 1”, “Type 2”) by their gene expression in the dorsolateral prefrontal cortex and explored biological differences between the subgroups. The study included post-mortem tissue samples that were sequenced in multiple, publicly available gene datasets using different sequencing methods. To evaluate the role of inflammation, the expression of genes in multiple components of neuroinflammation were examined: complement cascade activation, glial cell activation, pro-inflammatory mediator secretion, blood–brain barrier (BBB) breakdown, chemokine production and peripheral immune cell infiltration. The Type 2 schizophrenics showed widespread abnormal gene expression across all the neuroinflammation components that was not observed in Type 1 schizophrenics. Our results demonstrate the importance of separating schizophrenic patients into their molecularly defined subgroups and provide supporting evidence for the involvement of the immune-related pathways in a schizophrenic subset.

## 1. Introduction

Many recent studies have provided evidence that schizophrenia may be etiologically heterogeneous and can be divided into at least two classes with different pathophysiological signatures via behavioral and cognitive symptoms [1,2], cortical structure [3,4], gene expression patterns [5,6,7,8], and serum inflammation markers [9,10,11,12]. Despite this, most studies continue to evaluate schizophrenics as a pooled group [13,14,15]. Here, we briefly introduce our hypothesis that schizophrenics can be categorized into two molecularly distinct subtypes defined by the presence of neuroinflammation in the dorsolateral prefrontal cortex (DLPFC).

Schizophrenic patients are characterized by the presence of hallucinations, delusions and/or disorganized speech sometimes with dysfunction in social behaviors or cognitive impairment. These can be generally classified into three broad categories: positive psychotic symptoms, negative symptoms, and cognitive symptoms. Some studies have divided schizophrenics into two groups based on whether they primarily experience negative symptoms [1,2]. In comparison to other schizophrenics, those with more negative symptoms exhibit more widespread cortical volume decreases [16,17,18]. However, other neuroanatomical studies have identified subtypes based on brain structure differences not otherwise detectible by clinical phenotyping [3,4,19].

The biologic etiology of schizophrenia remains unclear, and the clinical heterogeneity associated with this disorder may have frustrated attempts to gain insights into the pathophysiology. Research in schizophrenia has shifted to determining if there are molecularly distinct subtypes. A recent study revealed two such subtypes in the DLPFC [6]. One subgroup (“Type 1”) has a DLFPC transcriptome indistinguishable from neurotypical controls. The other subgroup (“Type 2”) has a radically different transcriptome with over 3000 differentially expressed (DE) genes compared to controls. This suggests that schizophrenics can not only be divided by their symptoms and cortical structure, but by their gene expression profile as well. However, it remains unclear what the underlying biological differences are between the two genetic subtypes.

Several studies have provided evidence that inflammation and the immune system may play a role in the schizophrenia pathophysiology (e.g., [20,21,22,23,24]). Building on that, multiple research groups have identified a “high inflammation” or “immuno-subtype” where a subset of schizophrenics show upregulated expressed in immune or inflammation related genes [7,9,10,11,12,25,26]. A majority of these studies use peripheral blood markers to separate schizophrenics into “high inflammatory” and “low inflammatory” before investigating gene expression [9,10,11,12,26]. These groups examine different aspects of inflammation (e.g., cytokines, complement cascade, cell adhesion molecules). Only one study that we are aware of [7] has clustered gene patterns within the brain to identify a “high inflammation” schizophrenic group rather than using peripheral blood markers. There is substantial evidence that inflammation is present in a schizophrenic subgroup, and it will be important to understand if this pattern is consistent in additional neuronal transcriptome datasets.

In this study, we hypothesized that the distinct molecularsubtypes, Type 1 and Type 2, distinguished by the number of differentially expressed genes, would be also identified in additional gene expression datasets. We expected Type 2 but not Type 1 schizophrenics to have increased inflammation and immune system expression in comparison to controls. We performed a comprehensive investigation across well-known components of neuroinflammation including complement cascade activation, glial cell activation, pro-inflammation mediator secretion, blood–brain barrier (BBB) breakdown, chemokine production and peripheral immune cell infiltration. We also noted correspondence with the “low inflammation” and “high inflammation” groups identified by other studies. Overall, this study provides additional support to the hypothesis that neuroinflammation is prominent in a subset of schizophrenic patients and highlights the importance of separating subjects into biotypes. As such, future researchers should molecularlysubtype subjects to potentially gain a clearer picture of which treatments will be most effective for an individual patient.

## 2. Materials and Methods

### 2.1. Sources of Data

Postmortem brain tissue was obtained primarily from by the Offices of the Chief Medical Examiner’s Office of the District of Columbia and the Commonwealth of Virginia with informed consent from the legal next of kin. This tissue is used in all four of the following datasets.

For the two RNAseq datasets, the tissue collection and data generation were conducted by the Lieber Institute for Brain Development and the BrainSeq Consortium. Both Poly-A RNA (dataset 1, “polyA”) and RiboZero RNA (dataset 2, “riboZ”) were prepared from the DLPFC (and hippocampus) and those methods have been previously described in papers reporting that work [14,15].

For the Illumina microarray expression data (“Illumina”), the methods have been previously published [27,28]. Tissue was extracted from the DLPFC (and hippocampus) and Illumina HumanHT-12 v4 expression array data were generated according to the manufacturer’s protocols. This dataset is available at dbGaP study accession phs000979.v1.p1 (accessed 3 January 2017).

For the CMC RNAseq dataset (“CMC”), Brodmann areas 9 and 46 of the DLPFC were dissected from the frozen slabs and RiboZero RNA was prepared. This dataset is publicly available in Release 3 at https://www.synapse.org/#!Synapse:syn2759792/wiki/ (accessed 6 July 2020).

### 2.2. Transcriptome Analysis

For differential expression analysis, genes that had an average count less than the 15% quantile were removed and only schizophrenic patients and healthy controls over the age of 25 were included (PolyA: 183 controls, 168 schizophrenic patients; RiboZ: 194 controls, 142 schizophrenic patients; Illumina: 206 controls, 189 schizophrenic patients). Across these three datasets, there were 107 schizophrenic patients that appeared in each For the CMC dataset, we restrict to subjects that were in the other three datasets, this resulted in 27 schizophrenic and 22 controls. The R libraries Voom and Limma were used for differential expression analysis. The analysis was performed with age, diagnosis, sex, race, and RIN included as covariates. After differential expression analysis, the *p*-values were adjusted for multiple testing using the Benjamini-Hochberg procedure.

The count data was restricted to schizophrenic patients and genes with statistically significant differential expression. The data were transformed using the variance stabilizing transformation (VST) from the DESeq2 package. Following this, the VST data was adjusted for covariates (age, sex, race, and RIN) using the removeBatchEffect function from limma.

### 2.3. Cluster Analysis

K-means clustering from the factoextra package was used on the adjusted VST data to identify two clusters in both RNAseq datasets. GO ontology (GO) enrichment analysis was performed using enrichGO on each schizophrenic subtype’s differentially expressed genes.

### 2.4. Type 1 vs. Type 2 Differentiation

A differential expression analysis was run a second time. This time each of the schizophrenic subtypes were analyzed separately to identify the genes differentially expressed in the DLPFC. The number of differentially expressed genes is used to label the two types with “Type 1” schizophrenics having less differentially expressed genes than the “Type 2” schizophrenics.

### 2.5. Gene List Creation

PolyA and riboZ were preliminarily clustered using all the differentially expressed genes. Focusing on genes that were identified as differentially expressed in Type 2 schizophrenics, we compared the expression between Type 1 and Type 2 schizophrenics. Genes that showed a significant expression level difference (*t*-test *p*-value < 0.05) between Type 1 and Type 2 were selected. This procedure was carried out for the polyA, riboZ and Illumina dataset. We identified the genes that were reoccurring across the three datasets, which culminated in a gene list of 149 genes (see Appendix A) that showed differences in expression between Type 1 and Type 2 schizophrenics. All four datasets (polyA, riboZ, Illumina, and CMC) were restricted to this gene list, and then subsequently analyzed using the pre-existing clustering pipeline.

## 3. Results

### 3.1. Investigation of Replicabilty of Two Schizophrenic Subtypes

We developed a list of 149 genes that were shown to be influential in the separation of Type 1 and Type 2 subjects in preliminary studies (see Methods and Materials for more details). Each dataset was restricted to this list of 149 genes, and then clustered using K Means clustering. Cluster analysis of all the individuals for each dataset led to two distinct groups with different genec expression patterns (Figure 1A,B). This trend was not observed in controls (Appendix A). Comparison of the demographics between the two subtypes showed no significant differences or that a particular demographic is driving the separation (Figure 1B,D). In Bowen et al. (2019) [6], it was observed that these subtypes were distinguished by the number of differentially expressed genes. In dataset 1 (“polyA”), we observed 8 differentially expressed genes in “Type 1” compared to controls and 7584 differentially expressed gene for “Type 2”. We see a similar trend in the second dataset (“riboZ”), with 228 differentially expressed genes in “Type 1” and 6084 genes in “Type 2”. This reaffirms previous findings that there is one schizophrenic subtype with a DLPFC transcriptome similar to controls (Type 1), and one with a radically different one (Type 2).

We also used the gene list to cluster the Illumina dataset (accessed on 3 January 2017) and then compared the cluster assignments across polyA, riboZ, and the Illumina dataset. In total, there are 107 subjects that are shared across all three datasets. We see that in pairwise comparisons, there are approximately 75–80% of subjects that are consistently classified as either Type 1 or Type 2 (Figure 2A).

Across all three datasets, we identified that 45 subjects consistently clustered as Type 1 and 25 subjects as Type 2. There were 37 patients that were not, and we refer to these subjects as “Mix”. The Mix population may imply that we are capturing a continuum rather than static subtypes. There is also the possibility that Mix subjects are evidence of additional types that we are not aware of. These 107 subjects, whom we will call “consistently clustered”, and their respective subtypes will be used for the rest of the paper.

We also tested in a fourth and separate dataset, the CMC dataset, whether using the gene list would cluster the subjects similarly to the other three datasets. While this dataset was much smaller (*n* = 49 subjects, 22 Controls, 27 Schizophrenics), we found that 93% of the consistently clustered Type 1 and Type 2s were identically clustered in the CMC dataset (Figure 2B). We also took note that the datasets that used the same sequencing libraries, polyA and Illumina (used polyA library sequencing), and riboZ and CMC dataset (used riboZ library sequencing), had the highest rate of consistency. This could imply that using the same sequencing library is an important part of replicating gene expression findings.

For the consistently classified subjects, we investigated whether other demographic factors could have influenced the groups. For the age of schizophrenia onset, medication usage, cotinine/nicotine, and RNA preparation, it is unlikely that these were responsible for the SCZ clusters and subtypes (see Appendix A). Given this evidence, we choose to continue examining polyA and riboZ.

For all consistently classified SCZ groups, we calculated the number of differentially expressed (DE) genes compared to controls for polyA and riboZ, respectively. Type 1 schizophrenics in both datasets had no differentially expressed genes. We examined the number of DE genes that were shared between the SCZ groups. SCZ pooled, Mix and Type 2 had over 1000 genes in common in both datasets (Figure 3A,B). It is also notable that there are genes that are differentially expressed in SCZ pooled that are not differentially expressed for the other subtypes. This emphasizes that importance of identifying separating schizophrenics to study biologic mechanisms and pathophysiology.

We observe that there are several Type 2 DE genes with large log fold change (LogFC) values in comparison to the other groups for both datasets (Figure 3B,D). The overexpression of genes in Type 2 is not surprising, as we see this pattern in Figure 1A,B. Type 2 has over 200 genes that have a LogFC over 1 (double the expression compared to controls). Refer to Appendix A for a venn diagram of high LogFC genes.

Having identified two consistent schizophrenic subtypes, Type 1 and Type 2, we wanted to investigate whether either of these biotypes showed gene expression patterns indicative of neuroinflammation.

### 3.2. Neuroinflammation-Related Gene Expression in Type 1 and Type 2 Schizophrenics

Neuroinflammation is a complex response that involves the activation of the complement system and glial cells and secretion of proinflammatory mediators leading to blood–brain barrier breakdown and infiltration of peripheral immune cells. While a successful neuroinflammation response will eliminate invading pathogens and promote wound healing and angiogenesis, chronic inflammation can have detrimental effects that have been observed in many neurodegenerative diseases such as multiple sclerosis, Alzheimer’s disease and even autism. We examine whether either the Type 1 or Type 2 groups exemplify the typical indicators of increased neuroinflammation.

#### 3.2.1. Activation of Complement Cascade System

The complement system is an integral component of the innate immune system. The principal function of this system is to protect the host from infection by clearing debris, tagging pathogens for engulfment or destruction, and enhancing inflammation [29,30]. However, uncontrolled activation of the complement system can lead to massive release of inflammatory factors, phagocytosis, and induction of BBB damage [31]. It has been implicated in a variety of conditions spanning from autoimmune diseases to traumatic brain injury [29,31].

We observed that Type 2 schizophrenics have significantly overall higher expression in complement cascade genes compared to the other SCZ groups (PolyA: F(15.89), T2 vs. All Tukey HSD *p*-value < 0.05; RiboZ: F(18.95), T2 vs. All Tukey HSD *p*-value < 0.01; Figure 4A,C). Many complement cascade genes were differentially expressed in Type 2 schizophrenics (Figure 4B,D) and this effect was not observed for Type 1 or Mix. Specifically, *C4A* and *C4B* expression was increased in Type 2 schizophrenics, with logFC levels 5–12% higher than Type 1 or Mix (Figure 4B,D–F).

#### 3.2.2. Glial Cell Activation

Given that Type 2 schizophrenics demonstrated an increase in complement genes, we next decided to analyze genes related to the two prominent glial cells in neuroinflammation: microglia and astrocytes. Microglia are the resident macrophages in the central nervous system (CNS). When in their “resting” state, microglia can be involved in neurogenesis, neuroprotection and synaptic pruning [32]. When microglia become “activated”, they can produce proinflammatory cytokines and increase phagocytic activity. Astrocytes are the other glial cell present in the CNS and have two types that are present in grey matter and white matter. When an insult to the CNS occurs, astrocytes will change their structure and morphology and begin producing cytokines and complements, leading to scar formation, and often referred to as reactive astrogliosis [32].

For microglia, we examined an array of genes associated with microglial activation. Across both datasets, we observe significant logFC decreases in *P2RY12*, *P2RY13* and *CX3CR1* and a significant increase in FTL (ferritin) (Figure 5A,B). Both FCER1G and *CIITA* are also increased but did not survive correction for multiple testing in the riboZ dataset (Figure 5A,B). We also examined markers specific to M1 and M2 microglia. We observe that in M1 markers, *FCGR2A/CD16* (PolyA: F(7.819), T2 vs. All Tukey HSD *p*-value < 0.01; RiboZ: F(7.381), T2 vs. All Tukey HSD *p*-value < 0.01) and *FCGR3A/CD32* (PolyA: F(4.383), T2 vs. All Tukey HSD *p*-value < 0.05; RiboZ: F(6.163), T2 vs. All Tukey HSD *p*-value < 0.01), Type 2 have significant upregulation compared to Type 1, Mix and controls (Figure 5C–F). However, for the M2 phenotype marker *MRC1/CD206*, there is no observed difference (Figure 5G,H).

For astrocytes, we examined both maturity markers as well as astrogliosis markers. *VIM* is a marker for immature astrocytes, and we only see significant increases for Type 2s in polyA (PolyA: F(13.08), T2 vs. T1 and T2 vs. Control Tukey HSD *p*-values < 0.01; RiboZ: F(1.246), T2 vs. all Tukey HSD *p*-value > 0.05; Figure 6B). We observe that Type 2 schizophrenics do have significant increase in *GFAP* expression, more so in polyA (Figure 6B,C), Overall, there is significant upregulation in astrogliosis in Type 2 schizophrenics compared to the other groups (PolyA: F(6.663), T2 vs. All Tukey HSD *p*-value < 0.05; RiboZ: F(6.181), T2 vs. All Tukey HSD *p*-value < 0.05; Figure 6E,F). On closer examination of the individual astrogliosis genes, we see that most of these genes are differentially expressed in Type 2 with logFCs greater than 1 (Figure 6G,H). *SERPINA3* has the largest upregulation in Type 2 schizophrenics, with a logFC greater than 4 with a 10–26% increase compared to Type 1 schizophrenics (Figure 6G,H).

#### 3.2.3. Secretion of Pro-Inflammatory Mediators

Pro-inflammatory cytokines and mediators contribute to the amplification and maintenance of inflammation [33,34,35]. We investigated the expression of pro-inflammatory cytokines (*IL-18, 1L1B, IL-33, IL-6*), receptors (*IL1R1, IL6ST, TNFRSF1A/TNFR1*) and other mediators (*NFKB1, PTGS2, S100B*). Overall, there is increased consistent upregulation of the pro-inflammatory mediators in Type 2 compared to the other subtypes in both datasets (PolyA: F(4.84), T2 vs. all Tukey HSD *p*-value < 0.05; RiboZ: F(3.899), T2 vs. T1 and T2 vs. Mixl Tukey HSD *p*-value < 0.05; Figure 7A,B). When examining schizophrenics as a pooled group, there are minimal increases in some pro-inflammatory cytokines, like *IL-6*¸ but with logFCs less than 1 (Figure 7C,D). However, when separated into their biotypes, Type 2 schizophrenics had consistent upregulation in multiple pro-inflammatory cytokines (1L1B and IL-6) and receptors (*IL6ST, IL1R1, TNFRSF1A/TNFR1*) (Figure 7E,F). *IL-6* was the most significantly upregulated pro-inflammatory mediator, being increased by 10% and 12% logFC in Type 2 schizophrenics compared to Type 1 schizophrenics, for polyA and riboZ, respectively.

#### 3.2.4. Brain Microvascular Endothelial Cells (BMEC) and BBB Breakdown

There has been increasing interest in role of microvascular dysfunction and BBB disruption in schizophrenia. The BBB is composed of a layer of brain microvascular endothelial cells (BMEC), which interact with neurons, glial cells, and the extracellular matrix to form the neurovascular unit. BMECs play a role in BBB permeability, brain parenchymal nourishment, and immune privilege in the brain [36].

We examined a range of paracellular genes (*CLDN5*, *OCLN*, *ESAM*, *CDH5*), transcellular proteins (*ICAM1*, *VCAM1*), transporter proteins (*ABCC1*, *ABCBC1*, *ABCG1*), transmembrane proteins (*IFITM1*, *IFITM2*) and extracellular matrix proteins (*COL4A1*, *COL4A2*, *FN1 and LAMA*). There is a significant increase in expression across all BMECs in Type 2 schizophrenics compared to the other subtypes (PolyA: F(8.57), T2 vs. all Tukey HSD *p*-value < 0.05; RiboZ: F(7.24), T2 vs. all Tukey HSD *p*-value < 0.05; Figure 8A,B). We observe consistent increases in Type 2 schizophrenics in *ICAM1*, *IFITM1*, *IFITM2*, *CDH5*, *CLDN5*, *COL4A1*, and *COL4A2* (Figure 8C,D). *ICAM1* and *IFITM2* have the largest logFC in Type 2 schizophrenics being increased by 7–24% compared to Type 1 schizophrenics. *ABCG2* is significantly decreased in Type 2 compared to Type 1 (PolyA: −5%; RiboZ: −28%) which is replicated in both datasets (Figure 8C,D).

#### 3.2.5. Chemokine and Leukocyte Chemotaxis

Chemokines are an integral part of inflammation development and mediate various immune cell responses such as chemotaxis and immune activation. Activated microglia and astrocytes can produce many chemokines which will recruit peripheral immune cells to site of inflammation in the CNS. Outside of immune responses, chemokines are important for ensuring crosstalk between neurons, glial cells and peripheral immune cells ensuring normal brain function. We examined the mRNA expression of four chemokines important in the neuroinflammation response: *CXCL1* (PolyA: F(28.5), T2 vs. All Tukey HSD *p*-value < 0.05; RiboZ: F(19.5), T2 vs. All Tukey HSD *p*-value < 0.05), *CXCL2* (PolyA: F(13.84), T2 vs. All Tukey HSD *p*-value < 0.05; RiboZ: F(9.855), T2 vs. All Tukey HSD *p*-value < 0.05), *CXCL8*/*IL-8* (PolyA: F(8.377), T2 vs. T1 and T2 vs. Control Tukey HSD *p*-value < 0.05; RiboZ: F(4.868), T2 vs. All Tukey HSD *p*-value < 0.05), and *CCL2*/*MCP-1* (PolyA: F(11.69), T2 vs. All Tukey HSD *p*-value < 0.05; RiboZ: F(8.042), T2 vs. All Tukey HSD *p*-value < 0.05) [37,38]. All of which demonstrated increased mRNA expression levels in Type 2 schizophrenics compared to controls and the other SCZ subtype. However, CXCL8 was not significantly different between the Type 2 and Mix subgroup in the polyA dataset. (Figure 9A–D).

#### 3.2.6. Peripheral Immune Cell Infiltration

The combination of a permeable BBB and chemokine presence can create an opportunity for peripheral immune cell recruitment and penetration into the CNS. Leukocyte migration into the brain is often seen in other autoimmune and neurodegenerative disorders such as multiple sclerosis, Alzheimer’s Disease, and Parkinson’s Disease [39].

We examined a range of macrophage, monocyte, natural killer, and neutrophil markers to identify if any are abnormally expressed in the schizophrenic subtypes. Overall, we see that Type 2 schizophrenics have the largest absolute logFC compared to Pooled, Type 1 and Mix (PolyA: F(7.366), all Tukey HSD *p*-value < 0.05; RiboZ: F(8.046), all Tukey HSD *p*-value < 0.05; Figure 10A,B). Among individual markers, *CD14*, *CD163*, *CD93*, *FCGR1A*/*CD64*, *FPR1*, *S100A8*, *S100A9*, and *S100A12* are significantly increased in Type 2s compared to the other schizophrenic groups (Figure 10C,D). *S100A8*, *S100A9*, and *S100A12* are 8–34% upregulated in Type 2 compared to Type 1 schizophrenics. There is a small decrease in *ITGAX* that passes the multiple test correction (Figure 10C,D).

## 4. Discussion

We have demonstrated that in multiple genomic datasets, spanning varying sequencing libraries and methods, schizophrenic patients can be stratified into two subtypes using a set list of 150 genes (Figure 1 and Figure 2, Appendix A). One group, Type 1, has DLPFC transcriptome-like controls, while the other, Type 2, has a radically different transcriptome profile from controls. There were also subjects that seem to have similarities to both Type 1 and Type 2, we referred to these subjects as “Mix” supporting previous findings [6].

These results emphasize the importance for molecular studies to evaluate schizophrenics as a heterogenous population and consider parsing the schizophrenics into separate biotypes. We demonstrated that there are hundreds of genes that are differentially expressed in schizophrenics as a whole group that are not when evaluating the schizophrenic subtypes independently and vice versa (Figure 3). This may provide some insight into some of the inconsistency between studies regarding what genes are differentially expressed in schizophrenia [40]. Some of the inconsistency could also be derived from sequencing parameters such as the sequencing library, which our results suggest play a contributing role in replicating patient clusters and should be taken into consideration for future studies (Figure 2). Being able to consistently distinguish schizophrenic patients will have important implications for future studies to examine the effectiveness of different treatments for each type.

Based on the evidence that demonstrates schizophrenics can be consistently separated into two subtypes from the DLPFC transcriptome, we next investigated whether these subtypes correspond to the “low” and “high” inflammation biotypes observed by previous studies using peripheral cytokine expression [9,10,11,12,25,26]. It is important to note that the gene list used to define the Type 1 and Type 2 subgroups only contained 3 inflammatory-related genes (*IFITM2, IFITM3, and TMEM119*). Therefore, if increased inflammation is present in one of these subtypes, it is unlikely to be an artifact of the clustering parameters. To do this, we evaluated the gene expression across integral components of neuroinflammation: complement system activation, glial cell activation, pro-inflammatory mediator secretion, chemokine and leukocyte chemotaxis and peripheral immune cell infiltration into the CNS; these will be discussed in turn.

### 4.1. Activation of Complement Cascade System

We observed significant upregulation of complement cascade genes in Type 2 schizophrenics, specifically *C4A* and *C4B* (Figure 4), which supports previous research that the complement system is upregulated in a subset of schizophrenics [7]. However, Bowen et al. (2019) [6] did not observe any increases in complement cascade 4 expression despite segregating their data into schizophrenic subtypes. This could be attributed to the lack of an appropriate Illumina probe since there is only one probe that maps to *C4A* (ILMN_2179533) but also maps to *C4B* and *C4B_2* [41].

Uncontrolled complement activation is a key contributor to neuronal loss, local inflammation and BBB injury, often observed in chronic neurodegenerative and acute neuroinflammatory central nervous system conditions [31]. Complement cascade 4 loci, *C4A* and *C4B*, have been identified as overexpressed in schizophrenic patients and linked to abnormal synaptic pruning, reduced cortical synapse density, increased microglial engulfment of synapses and altered social behavior [30,42,43,44,45,46]. Our results demonstrate an upregulation in the complement cascade in Type 2 schizophrenics, implicating there may be subsequent exacerbation of neuroinflammation, aberrant synaptic pruning and tissue damage.

### 4.2. Glial Cell Activation

We identified decreases in microglial markers *P2RY12*, *P2RY13* and *CX3CR1* in Type 2 schizophrenics (Figure 5A,B), corroborating findings from previous studies [11,47,48]. Decreases in *P2RY12*, *P2RY13* and *CX3CR1* may lead to altered neuron-microglial signaling in Type 2s via aberrant synaptic pruning, phagocytosis, and dopamine release [11,49,50,51,52]. *FTL*, *FCGR2A*/*CD16*, and *FCGR3A*/*CD32* were increased in the Type 2 group (Figure 5A,C–E). *FTL* is linked to iron retention in microglia, and generally increases in response to microglia activation [33,53]. However, other microglia activation markers, such as IBA-1/AIF-1, were not increased (Figure 5A,B), perhaps suggesting that there is a subdued microglial response. *FCGR2A*/*CD16* and *FCGR3A*/*CD32* are M1 microglial phenotype markers, indicating an increased number of M1 microglia which are associated with inflammation, cytotoxicity, and acute immune response [33].

We observed widespread increases in astrogliosis gene expression in Type 2 schizophrenics compared to the other groups (Figure 6C,D). *SERPINA3* has been consistently increased in schizophrenic post-mortem brain tissue [9,54,55], and could possibly be a marker of persistent reactive gliosis [56]. Other astrogliosis genes are important in the inflammatory response (OSMR), oxidative stress response (CP, FBLN5), and blood–brain barrier integrity (IFITM3) [12]. Astrogliosis can both protect healthy tissue and prevent brain damage repair via inhibition of axon regeneration and synaptogenesis and their role may depend on the subtype of astrocyte [12,56]. Therefore, more work will need to be conducted to evaluate the role of astrogliosis and the prevalence of astrocyte subtypes in Type 2 pathology.

Our data suggests that there may be a subdued microglial response and a strong astrogliosis response in Type 2 schizophrenics. This appears to be a paradoxical situation; however, reactive astrocytes have been proposed to release factors that keep microglia in a non-inflammatory state contributing to a prolonged pro-inflammatory response produced by other cells [12,57]. It is also possible that a chronic activated astroglial state is contributing to the aberrant synaptic pruning by microglia [57]. Therefore, inflammatory astrocytes and non-inflammatory microglia could mutually be contributing to neuroinflammation through their effects on each other.

### 4.3. Secretion of Pro-Inflammatory Mediators

Increased expression of pro-inflammation markers in schizophrenic patients has been previously reported by several studies, particularly in peripheral blood samples and serum [20,58,59]. Type 2 schizophrenics had consistent upregulation in pro-inflammatory cytokines, *1L1B* and *IL-6*, and inflammation receptors, *IL1RL1*, *IL6ST*, and *TNFRSF1A*/*TNFR1*, which supports previous findings from “high inflammation” subgroups [9,10,11]. In contrast to these previous studies, we used brain mRNA expression to subdivide the schizophrenics and not peripheral cytokine expression. However, there has been reported concordance between the expression levels of inflammation-related genes in the periphery and in the frontal cortex [22].

Pro-inflammatory mediators are at the center of the neuroinflammation response. They are produced by activated microglia and astrocytes, leading to further activation of glial cells, breakdown of the BBB and production of chemokines. Chronic activation of pro-inflammatory mediators, particularly cytokines, can lead to neurodegeneration and have been linked to increased severity of clinical symptoms (e.g., negative symptoms) and increased PANSS scores [60,61]. Therefore, our findings of increased cytokines in Type 2 continue to support the hypothesis that neuroinflammation is prominent in a schizophrenic subgroup.

However, the source of these cytokines remains speculative and additional research will be needed to localize inflammatory cytokine transcripts to specific cells. Based on our identification of increased astrogliosis and reduced microgliosis, it is possible that astrocytes are a driver of cytokine production, particularly *IL-6* [61]. However, there have been conflicting studies that have no observed an increase in inflammation or pro-inflammatory mediators [13,14,62,63]. This could be attributed to the evaluation of schizophrenics as a single group. We demonstrated that while some of these mediators are differentially expressed in pooled schizophrenics, the logFC values are small and may not be representative of the heterogeneous expression in schizophrenics (Figure 4C,D). This highlights the importance of separating schizophrenics into subtypes, as examining schizophrenics altogether may dampen the differences in inflammation gene expression.

### 4.4. Brain Microvascular Endothelial Cells (BMEC)

Type 2 schizophrenics have increased expression in *ICAM1*, *IFITM1*, *IFITM2*, *CDH5*, *CLDN5*, *COL4A1*, and *COL4A2* and decreased expression in *ABCG2* (Figure 8C,D). *ICAM1* and *CDH5* contribute to the regulation of BBB permeability and are increased during inflammation in response to leukocyte migration [26,36,64]. Both have been observed to be increased in “high inflammation” schizophrenics [11,26]. *ABCG2*, an efflux transporter, has been previously identified as having decreased expression in “high inflammation” schizophrenics. [26]. Increases in *IL1B* and *IL-6* have been reported to reduce both mRNA and protein expression of ABCG2 in vitro [65], which would be consistent with our results. Decreases in *ABCG2* could increase the vulnerability of the BBB to harmful compounds [36,65]. Due to their role in BBB permeability, elevated *ICAM1* and *CDH5* and decreased *ABCG2* may imply that there is BBB breakdown allowing for leukocyte migration in Type 2 schizophrenic patients.

*CLDN5* is a major BBB component crucial for angiogenesis and endothelial maintenance that has been reported as increased [66], decreased [67] or unchanged [66,68] in schizophrenia. We are the first study among these to examine CLDN5 by schizophrenic subtype which may explain the differences between our results and previous studies. Additionally, discrepancy between the mRNA and protein expression of CLDN5 has been consistently reported potentially due to the breakdown of CLDN5 by cAMP signaling and downstream protein kinase A signaling [69].

The *IFITM* genes (*IFITM1*, *IFITM2*), which are found to be increased in a subset of schizophrenics, are expressed in brain endothelial cells and their function appears to be antiviral [26,70]. While it remains unclear what the source of elevated *IFITM* is in schizophrenia, there is some evidence that suggests maternal immune activation alone is not enough to upregulate IFITM [71]. However, upregulation of *IL-6*, *IL-8* and *IFITM* (Figure 7C,D and Figure 9C) may implicate a molecular cascade of immune activation in Type 2 schizophrenics [71]. This cascade can be a source of further exploration to directly quantify the levels of immune marker protein and their cell source.

### 4.5. Chemokine and Leukocyte Chemotaxis

Overall, we observed overexpression of four chemokines (*CXCL1*, *CXCL2*, *CXCL8*, *CCL2*/*MCP-1*) in Type 2 schizophrenics (Figure 9). *CXCL1*, *CXCL2*, and *CXCL8/IL-8* are contributors to granulocyte and leukocyte migration, particularly in other autoimmune diseases such as multiple sclerosis and experimental autoimmune encephalomyelitis [61,72,73,74]. Increases in *CXCL8/IL-8* have been observed in peripheral blood samples in pooled schizophrenics and mRNA expression levels in the DLPFC in “high inflammation” schizophrenics [9,75]. *CCL2/MCP-1* is known to recruit monocytes, macrophages, activated T cells and natural killer (NK) cells to damaged tissue in the CNS after trauma, infection, or toxin exposure [37,76]. *CCL2/MCP-1* has been observed to both be higher [77,78,79,80,81] and unchanged [82,83,84,85] in the plasma of schizophrenic patients compared to controls. In combination, increases in astrocyte activation, cytokine production, and BBB breakdown create an environment where upregulation in these chemoattractant signals may be inducing the migration of leukocytes through the BBB into the CNS and possibly contributing to exacerbated negative symptoms in Type 2 schizophrenics [61,74,75].

### 4.6. Peripheral Immune Cell Infiltration

Expression levels of macrophage markers, *CD163* and *FCGR1A/CD64*, and monocyte markers, *CD14* and *CD93* are increased in Type 2 schizophrenics but not the other schizophrenic biotypes (Figure 10C,D). CD163+ macrophages have been identified in brain tissue proximal to neurons in “high inflammation” schizophrenics as well as in the brain parenchyma [26]. Therefore, we provide preliminary evidence that macrophages and monocytes may be infiltrating the CNS in Type 2 schizophrenics. Further studies will be needed that can use specific markers to differentiate between brain-resident and blood-derived immune cells.

We also observed significant overexpression in markers associated with neutrophils (*FPR1*, *S100A8*, *S100A9*, and *S100A12*) in the Type 2 biotype. In healthy individuals, there are very little neutrophils within in the brain but neutrophil infiltration into the CNS is a well-documented phenomenon in various pathologies (e.g., infection, trauma, neurodegeneration or autoimmune) [39,73,86,87]. Previous studies have established links between schizophrenia and increased neutrophil counts in the periphery providing preliminary evidence that neutrophils may also be infiltrating the brain [25,88]. The transmigration of neutrophils to the brain has been linked to the depression-like behaviors and memory deficits during severe systemic inflammation [87,89]. We are unaware of any studies to date that have observed neutrophils in the CNS of schizophrenic patients so this may be a potential field of study in Type 2 schizophrenics.

### 4.7. Limitations of This Work

We acknowledge that our study is limited in its scope. Our data is restricted with the use of postmortem samples, and we have very minimal information on the clinical symptoms and features of the patients. While we do have a small amount of information on the individuals’ demographic information, we have no data on the environmental comorbidities that these individuals may experience (e.g., infections, trauma, stress). It is well documented that environmental stressors are correlated with inflammation ([20]). Additionally, while we observed increases in many inflammation-related mediators, we do not have any histological or immunostaining data that would pinpoint the source of these effectors.

We observe that for several genes, the Mix group had expression levels in between Type 1 and Type 2. It is possible that either these subtypes are not static, and we may be capturing the different schizophrenic stages (e.g., first episode psychosis vs. chronic) or these groups could also represent a continuum of inflammatory status (e.g., “low”, “medium”, and “high”). We also do not have any information on anti-inflammatory medications which could be contributing to the “Mix” group’s appearance of a reduced immune expression compared to Type 2s. Again, due to the limited information available regarding these patients, we are unable to conclusively identify whether we are detecting distinct subtypes or a continuum and if or how those relate to the patient’s clinical time course.

## 5. Conclusions

This study focused on investigating whether schizophrenic patients stratified into heterogeneous groups, and if neuroinflammation was a prominent feature of one of these types. We were able to consistently classify schizophrenics into Type 1, Mix and Type 2 biotypes. Through our comprehensive landscape analysis of neuroinflammation components, we’ve identified abnormal gene expression in Type 2 schizophrenics that is not observed in Type 1 or Mixed schizophrenics. Chronic neuroinflammation can have detrimental effects on healthy brain function leading to more cognitive deficits and increased negative symptoms. This research demonstrates that differentiating schizophrenics into subgroups can highlight critical differences, such as the presence of neuroinflammation. Additionally, evaluating these schizophrenic subtypes individually may explain why previous studies have shown inconsistent findings. Full exploration of schizophrenic heterogeneity may inform how patients are diagnosed and treated, especially as precision medicine and pharmacogenomics become more integrated into clinical care.

## Figures and Tables

**Figure 1 genes-13-01200-f001:**
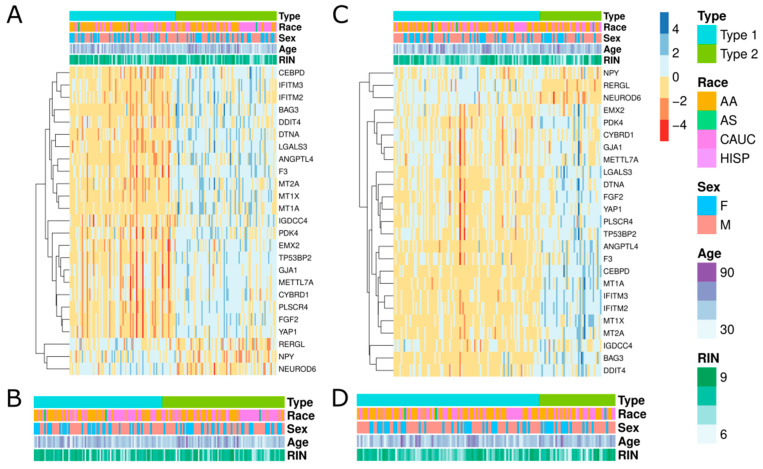
Two schizophrenic clusters identified in multiple RNAseq datasets. (**A**,**C**) Heatmaps for polyA and riboZ, respectively, showing the top forty genes with the highest log fold change (LogFC) across all schizophrenic patients. Clustering of the genes and subjects are shown at the side and top of the heatmaps, respectively. (**B**,**D**) Enlargement of the color bars in (**A**,**C**). Refer to the color bar on the side for color definitions. AA = African American, AS = Asian, CAUC = Caucasian, HISP = Hispanic, F = Female, and M = Male.

**Figure 2 genes-13-01200-f002:**
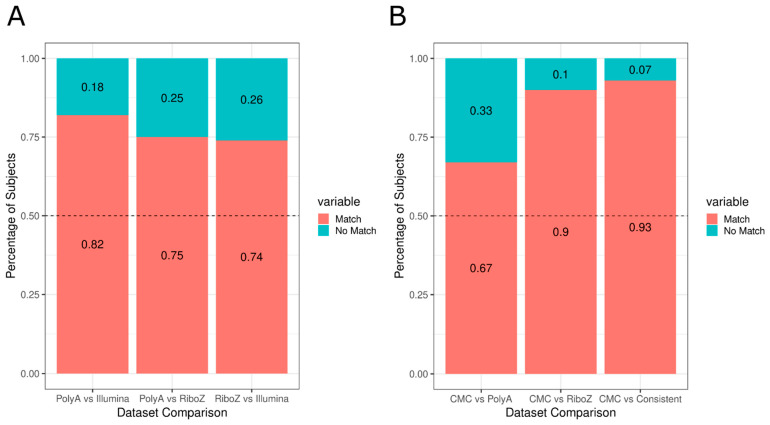
Most subjects are being consistently clustered across four independent datasets. (**A**) Comparison of polyA, riboZ and Illumina. (**B**) Comparison of polyA, riboZ, and consistently classified subjects from (**A**) with CMC. For each pair of datasets, the percentage of subjects that were clustered as the same type (“match”, pink) versus not the same subtype (“no match”, blue) was calculated. To see more detailed tables, refer to Appendix A.

**Figure 3 genes-13-01200-f003:**
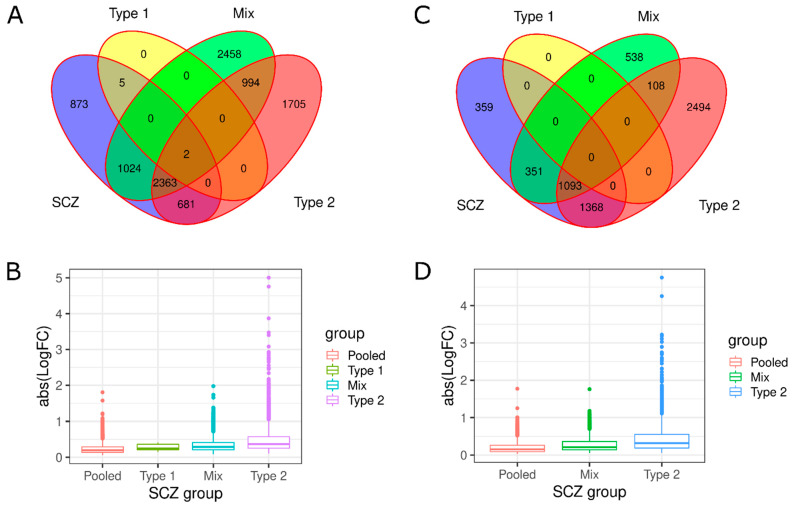
Type 2 schizophrenics have large differences in expression compared to schizophrenics (SCZ) pooled, Type 1, and Mix in both datasets. (**A**,**C**) Venn diagram of overlapping genes across the SCZ groups. Number of genes that are shared between groups will be listed in intersecting circles. SCZ pooled (blue), Type 1 (yellow), Mix (green) and Type 2 (red). (**B**,**D**) Boxplot of absolute log fold change (LogFC) values for each SCZ group in polyA and riboZ, respectively. The absolute value of LogFC was calculated.

**Figure 4 genes-13-01200-f004:**
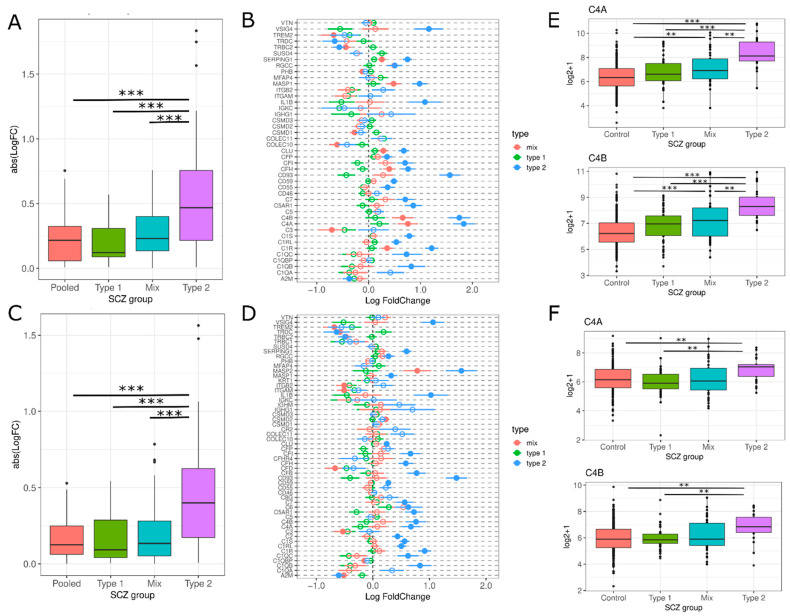
Type 2 has increased upregulation of complement cascade genes. (**A**,**C**) Absolute LogFC value for complement cascade genes for each SCZ group for polyA and riboZ, respectively *** *p*-value < 0.001 via Tukey HSD. (**B**,**D**) LogFC for each SCZ group for subset of genes in polyA and riboZ, respectively. Type 2 (blue), Mix (red), Type 1 (green). Lines indicate standard error. Filled circles = differentially expressed (Benjamini-Hochberg *p*-value < 0.05), empty circles = not differentially expressed. (**E**,**F)** Boxplot of the log2+1 expression data is shown for each SCZ group: Controls (red), Type 1 (green), Mix (blue) and Type 2 (purple). ** *p*-value < 0.01, *** *p*-value < 0.001 via Tukey HSD.

**Figure 5 genes-13-01200-f005:**
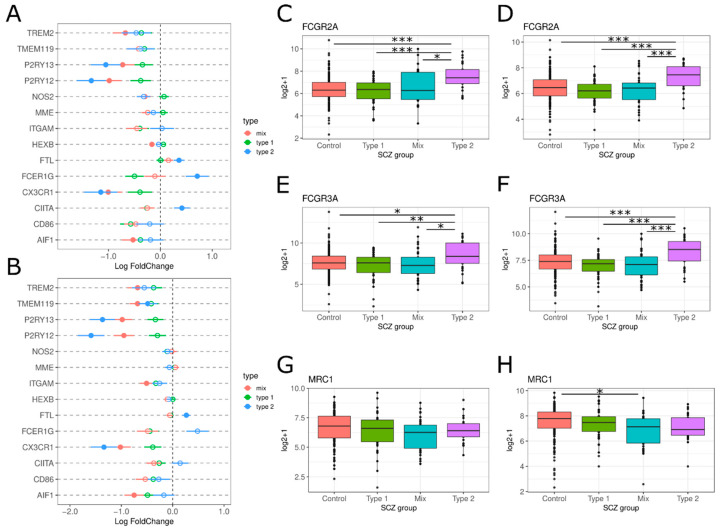
Microglia marker expression in the SCZ groups. (**A**,**B**) LogFC for genes associated with microglial activation across SCZ subtypes: Type 2 (blue), Mix (red), Type 1 (green). Lines indicate standard error. Filled circles = differentially expressed (Benjamini-Hochberg *p*-value < 0.05), empty circles = not differentially expressed. (**C**–**H**) Boxplot of the log2+1 expression data is shown for each SCZ group: Controls (red), Type 1 (green), Mix (blue) and Type 2 (purple). Graphs on the left are polyA and graphs on the right are riboZ. * *p*-value < 0.05, ** *p*-value < 0.01, *** *p*-value < 0.001 via Tukey HSD.

**Figure 6 genes-13-01200-f006:**
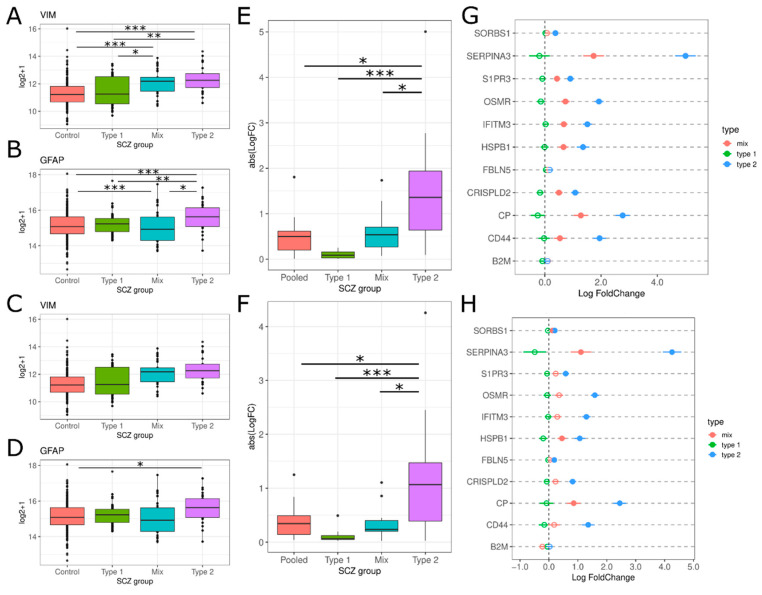
Astrocyte marker expression in the SCZ groups. (**A**–**D**) Boxplot of the log2+1 expression data is shown for each SCZ group: Controls (red), Type 1 (green), Mix (blue) and Type 2 (purple). (**A**,**B**) are from the polyA dataset, (**C**,**D**) for riboZ. * *p*-value < 0.05, ** *p*-value < 0.01, *** *p*-value < 0.001 via Tukey HSD. (**E**,**F**) Absolute LogFC value for astrocyte activation genes for each SCZ group for polyA and riboZ, respectively. * *p*-value < 0.05, *** *p*-value < 0.001 via Tukey HSD. (**G**,**H**) LogFC for genes associated with astrocyte activation across SCZ groups for polyA and riboZ, respectively: Type 2 (blue), Mix (red), Type 1 (green). Lines indicate standard error. Filled circles = differentially expressed (Benjamini-Hochberg *p*-value < 0.05), empty circles = not differentially expressed.

**Figure 7 genes-13-01200-f007:**
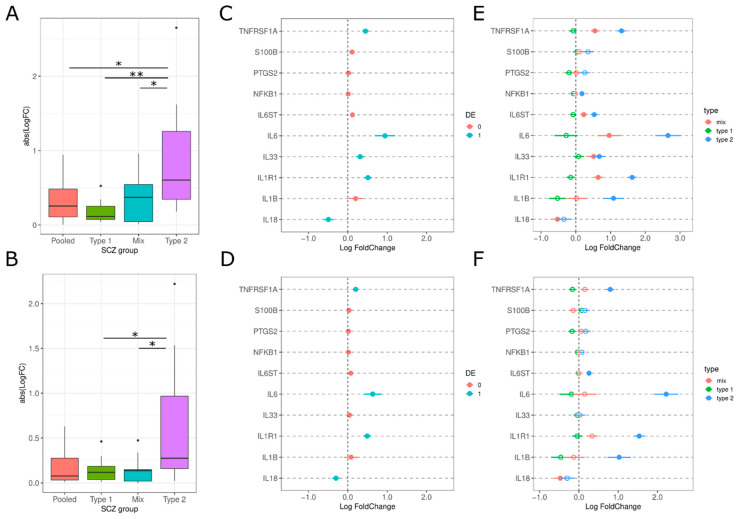
Pro-inflammatory mediator expression among SCZ subtypes. (**A**,**B**) Absolute LogFC value for astrocyte activation genes for each SCZ group for polyA and riboZ, respectively. * *p*-value < 0.05, ** *p*-value < 0.01, via Tukey HSD. (**C**,**D**) LogFC for pro-inflammatory mediators for SCZ as a pooled group. Lines indicate standard error. Pink circles = differentially expressed (Benjamini-Hochberg *p*-value < 0.05), blue circles = not differentially expressed. (**E**,**F**) LogFC pro-inflammatory mediators across SCZ subtypes for polyA and riboZ, respectively: Type 2 (blue), Mix (red), Type 1 (green). Lines indicate standard error. Filled circles = differentially expressed (Benjamini-Hochberg *p*-value < 0.05), empty circles = not differentially expressed.

**Figure 8 genes-13-01200-f008:**
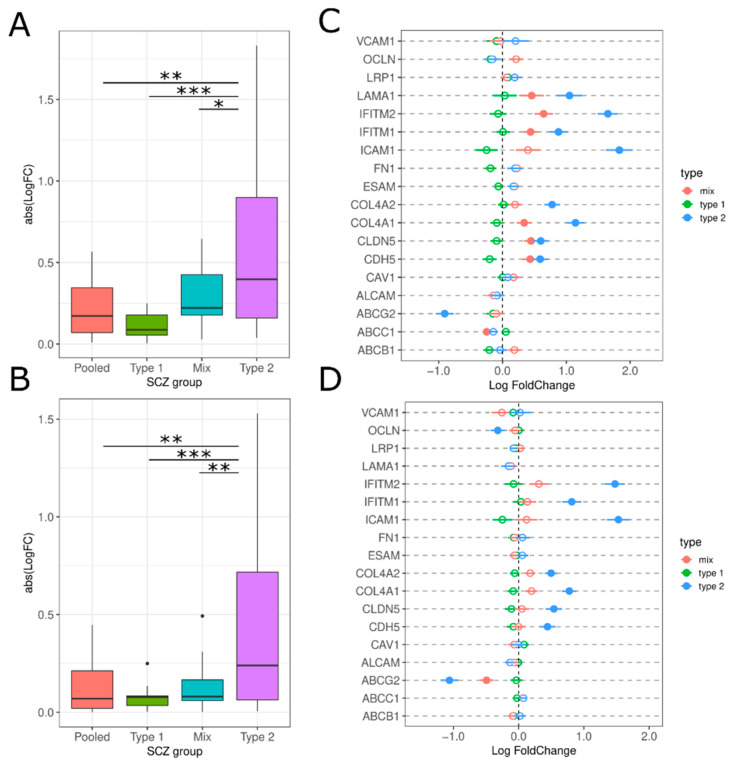
BMEC expression in the SCZ groups. (**A**,**B**) Absolute LogFC value for BMEC genes for each SCZ group for polyA and riboZ, respectively. * *p*-value < 0.05, ** *p*-value < 0.01, *** *p*-value < 0.001 via Tukey HSD. (**C**,**D**) LogFC for each BMEC gene across SCZ subtypes for polyA and riboZ, respectively: Type 2 (blue), Mix (red), Type 1 (green). Lines indicate standard error. Filled circles = differentially expressed (Benjamini-Hochberg *p*-value < 0.05), empty circles = not differentially expressed.

**Figure 9 genes-13-01200-f009:**
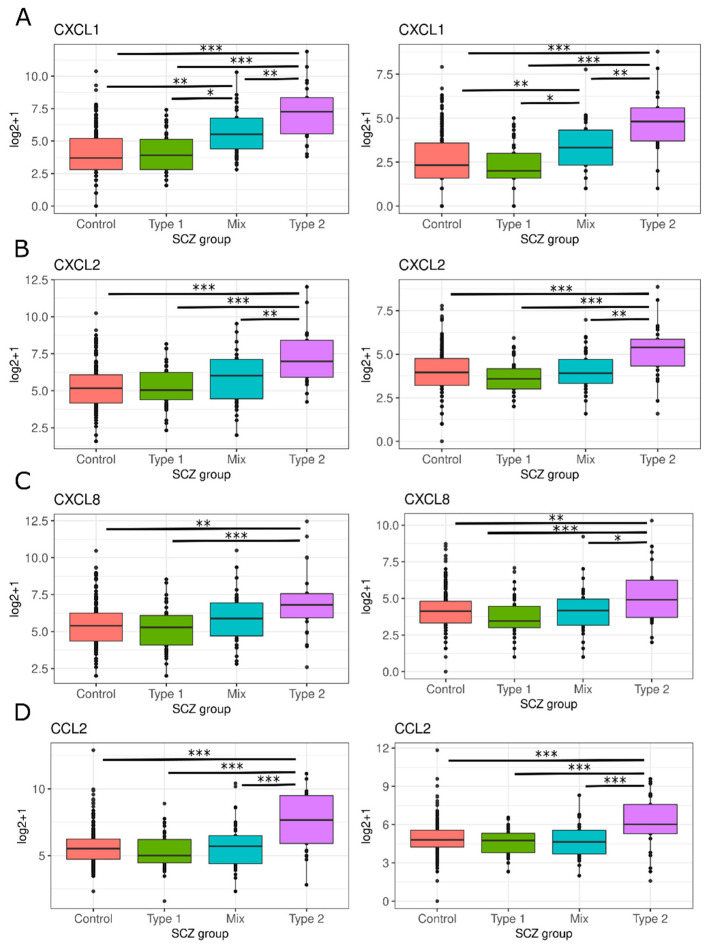
Chemokine’s expression across SCZ subtypes: (**A**) CXCL1, (**B**) CXCL2, (**C**) CXCL8, (**D**) CCL2. Boxplot of the log2+1 expression data is shown for each SCZ group: Controls (red), Type 1 (green), Mix (blue) and Type 2 (purple). Graphs on the left are polyA and graphs on the right are riboZ. * *p*-value < 0.05, ** *p*-value < 0.01, *** *p*-value < 0.001 via Tukey HSD.

**Figure 10 genes-13-01200-f010:**
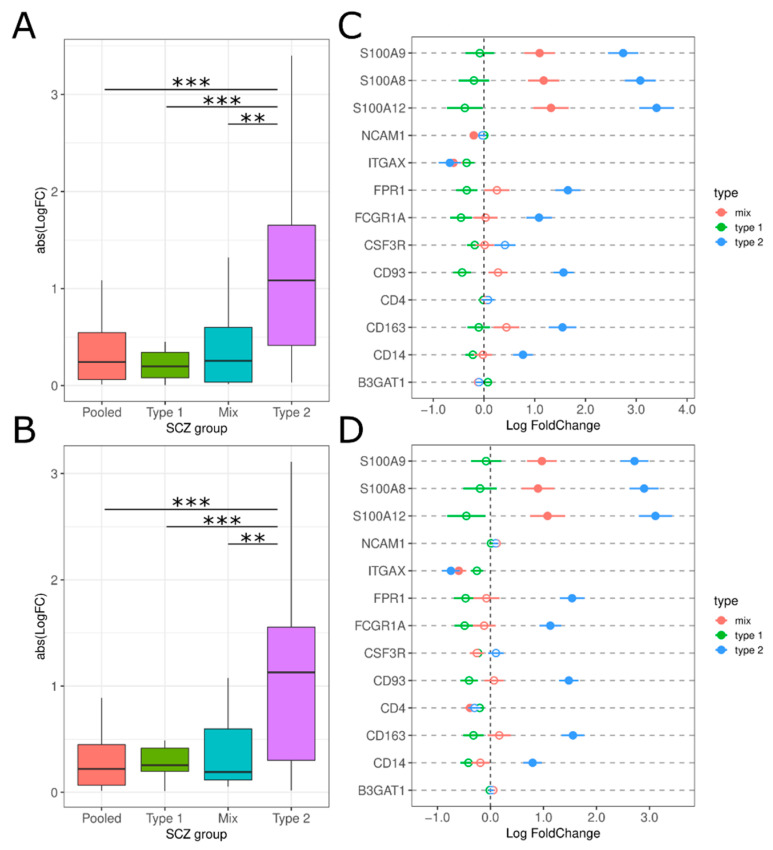
Peripheral immune expression in the SCZ groups. (**A**,**B**) Absolute LogFC value for peripheral markers for each SCZ group for polyA and riboZ, respectively. ** *p*-value < 0.01, *** *p*-value < 0.001 via Tukey HSD. (**C**,**D**) LogFC for each peripheral marker across SCZ subtypes for polyA and riboZ, respectively: Type 2 (blue), Mix (red), Type 1 (green). Lines indicate standard error. Filled circles = differentially expressed (Benjamini-Hochberg *p*-value < 0.05), empty circles = not differentially expressed.

## Data Availability

The RNAseq data presented in this study are openly available from the BrainSeq Consortium (polyA: http://eqtl.brainseq.org/phase1/ (accessed on 1 April 2020); riboZ: http://eqtl.brainseq.org/phase2 (accessed on 1 April 2020)/) and CommonMind Consortium (https://www.synapse.org/#!Synapse:syn2759792/wiki/) (accessed on 6 July 2020). The Illumina microarray is available at dbGaP study accession phs000979.v1.p1 (accessed on 3 January 2017).

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
