# Peer review of "Immune-Related Genomic Schizophrenic Subtyping Identified in DLPFC Transcriptome"

_genes, 2022, doi:10.3390/genes13071200_

Round 1
Reviewer 1 Report
The authors highlighted a subgroup of SZ patients (Type II) that showed differential gene expression in the inflammation pathway, as compared to other SZ patients (Type I) and controls. Type II SZ patients showed increased inflammatory phenotype, microglia/astrocytes anomalies and BBB-related impairments. The approach of gene expression analysis in Type I and II was hypothesis-driven, meaning that the authors focused only on inflammatory-related genes. Although this choice may be relevant as inflammation was widely found to play a role in the pathophysiology of SZ, this approach lack a bite of innovation. The pathway analysis regarding microglia/astrocytes, or cytokines, or BBB -related genes is of certain interest indeed, but it would be maybe worth to investigate other pathways that were found to be related to SZ, and may segregate in Type I or Type II, or even in the “Mix” group. With RNAseq, the huge amount of data allow more broad pathways analysis, which is more and more used nowadays in order to highlight subgroups of SZ, following the hypothesis of heterogenous disease. Therefore, it is a bit a pity that the authors stuck to this reductive hypothesis of inflammation, while a broader approach may be more relevant.
Overall, this study confirm the inflammation role in subgroups of SZ, which was already widely described. As the authors mentioned in their discussion, the limitation of postmortem studies is the effect of chronicity of the disease and the restricted amount of clinical data, which would relate this pathophysiological aspect to symptomatology.
Comments on the Methods:
- The number of individuals (controls and schizophrenia patients) used in each dataset is not mentioned in the Material and Methods section.
- In the first part of the results, the authors performed a preliminary analysis on all genes from the 2 RNAseq datasets to highlight the genes that were highly involved in the separation between Type I and II schizophrenia patients, but it is not clear on which base were defined Type I and II for the comparison. Is it on the previous paper (Bowen et al., 2019, transl psych)?
- The authors perform a K-mean analysis using a set of genes that separate the most Type I and II schizophrenia patients. They find that the cluster analysis reveals 2 groups that have a different profile of gene expression. I am a bit confused about the rational of the K-mean analysis: if the K-mean is performed using a set of genes that separate the most 2 groups, don’t you expect to find indeed 2 separated clusters after the K-mean? Also, the right classification of individuals to the correct cluster seems quite obvious, as the clusters were defined on the gene expression that separate the most these same individuals, isn’t it?
- Another point is the use of 149 genes to classify 107 subjects. When the number of parameters used in the cluster analysis is higher than the number of subject to classify, the model has higher risk of overfitting.
- What are the 149 genes that are differentially expressed between Type I and II, which were used to define these two clusters? Are the inflammatory-related genes present in this list? It would be interesting to know, whether the difference in inflammatory-related genes between type I and II is real and not artificially defined by the clustering…
Comments on the results:
- The authors find an upregulation of the complement related genes, especially C4A and C4B in Type II. It may be relevant to investigate the genetic architecture of the gene expression in both SZ Types, as performed in Sekar et al., (Nature, 2016), who found that some specific allele forms were associated with higher risk for SZ and induced higher expression of C4A. It would be interesting to know wether the Type II bear this genetic alleles that predispose to higher C4 expression, which would confirm the stratification approach of the authors, based on genetic risk factors.
Sekar A, Bialas AR, Rivera Hd, et al. Schizophrenia risk from complex variation of complement component 4. Nature. 2016;530(7589):177 - 183. doi:10.1038/nature16549
Comments on the discussion:
- In the “3. Secretion of Pro-Inflammatory Mediators” section, the authors compare their study to previous work investigating “high inflammation” subgroup in SZ patients. They refer to previous studies showing this subdivision only on peripheral material, in contrast to their study on brain mRNA. However, the authors should quote the study of Fillman and colleagues, that also found increased inflammation-related gene expression in the DLPCx of SZ patients and who also identified a “high inflammation” group, mostly represented by SZ patients. Noteworthy, the study of Fillman et al., find the same genes to be highly expressed as in the present manuscript, corroborating their results.
Fillman SG, Cloonan N, Catts VS, et al. Increased inflammatory markers identified in the dorsolateral prefrontal cortex of individuals with schizophrenia. Molecular psychiatry. 2012;18(2):206 - 214. doi:10.1038/mp.2012.110
General comments:
Legends of Figure 1 and 2 are the same. The legend of Figure 2 is missing?
Legends of Figure 5 and 6 were exchanged
Author Response
The authors highlighted a subgroup of SZ patients (Type II) that showed differential gene expression in the inflammation pathway, as compared to other SZ patients (Type I) and controls. Type II SZ patients showed increased inflammatory phenotype, microglia/astrocytes anomalies and BBB-related
impairments. The approach of gene expression analysis in Type I and II was hypothesis-driven, meaning that the authors focused only on inflammatory-related genes. Although this choice may be relevant as inflammation was widely found to play a role in the pathophysiology of SZ, this approach lack a bite of innovation. The pathway analysis regarding microglia/astrocytes, or cytokines, or BBB -related genes is of certain interest indeed, but it would be maybe worth to investigate other pathways that were found to be related to SZ, and may segregate in Type I or Type II, or even in the “Mix” group. With RNAseq, the huge amount of data allow more broad pathways analysis, which is more and more used nowadays in order to highlight subgroups of SZ, following the hypothesis of heterogenous disease. Therefore, it is a bit a pity that the authors stuck to this reductive hypothesis of inflammation, while a broader approach may be more relevant. Overall, this study confirm the inflammation role in subgroups of SZ, which was already widely described. As the authors mentioned in their discussion, the limitation of postmortem studies is the effect of chronicity of the disease and the restricted amount of clinical data, which would relate this pathophysiological aspect to symptomatology.
This paper is pointedly investigating whether the same distinction between two schizophrenic clusters reflects the same findings as previous studies, for example, the Type 1 and Type 2 difference and/or “low” and “high” inflammation. We use a targeted approach to answer these questions and a broad pathway analysis would not be an appropriate method. Additionally, for the consistently classified subjects, Type 1 schizophrenics had no differentially expressed genes compared to controls so additional brain regions would need to be investigated to fully assess Type 1s which is not in the scope of this paper.
Comments on the Methods:
The number of individuals (controls and schizophrenia patients) used in each dataset is not mentioned in the Material and Methods section.
Please refer to Section 2.2. Transcriptome Analysis (page 3, line 100) to see added details regarding the number of patients and controls for each dataset.
In the first part of the results, the authors performed a preliminary analysis on all genes from the 2
RNAseq datasets to highlight the genes that were highly involved in the separation between Type I and II schizophrenia patients, but it is not clear on which base were defined Type I and II for the comparison. Is it on the previous paper (Bowen et al., 2019, transl psych)?
The number of differentially expressed genes is used to label the two types with “Type 1” schizophrenics having less differentially expressed genes than the “Type 2” schizophrenics, as described in Section 2.4 (page 3, line 120).
The authors perform a K-mean analysis using a set of genes that separate the most Type I and II
schizophrenia patients. They find that the cluster analysis reveals 2 groups that have a different profile of gene expression. I am a bit confused about the rational of the K-mean analysis: if the K-mean is performed using a set of genes that separate the most 2 groups, don’t you expect to find indeed 2 separated clusters after the K-mean? Also, the right classification of individuals to the correct cluster seems quite obvious, as the clusters were defined on the gene expression that separate the most these same individuals, isn’t it?
It was not guaranteed that we would get consistent clustering even with our set of 149 genes. There has been incredible variability between studies regarding the expression of specific genes in schizophrenia (e.g. CLDN5). The four datasets we used all contained the same subjects but had different research groups, sequencing platforms and sequencing libraries. Therefore, it was a completely likely scenario that even using the same list of genes would result in different patient clusters simply due to dataset variability. So, the fact that we can get consistent clustering across four independent datasets is a relevant finding and can be used in future studies to further study the heterogeneity of schizophrenia.
Another point is the use of 149 genes to classify 107 subjects. When the number of parameters used in the cluster analysis is higher than the number of subject to classify, the model has higher risk of
overfitting.
The reviewer expresses an important concern. When a model has more degrees of freedom than
subjects, over-fitting is a high risk. Often, models have at least as many degrees of freedom as features
(genes). However, in this study, we are simply using K-Means to subdivide the schizophrenic population.
The number of degrees of freedom in a K-Means model is not a function of the number of genes*.
Therefore, this concern does not apply.
* https://www.sciencedirect.com/science/article/pii/S0167947320300657
What are the 149 genes that are differentially expressed between Type I and II, which were used to
define these two clusters? Are the inflammatory-related genes present in this list? It would be
interesting to know, whether the difference in inflammatory-related genes between type I and II is real and not artificially defined by the clustering
The full list of 149 genes is now included in Supplemental Table 1. We identified that only three genes
(IFITM2, IFITM3 and TMEM119) were inflammation-related and used in our landscape analysis.
Therefore, it is unlikely that our findings of increased neuroinflammation in Type 2 schizophrenics is an artifact of the clustering parameters. This is now included in the discussion in the third paragraph (page 14, line 415).
Reviewer 2 Report
an excellent and robust manuscript on a very important subject of differentiating different sub groups of schizophrenia. I see that the authors have attempted to show that some schizophrenic patients may be struggling with chronic inflammation and how important it is to differentiate such patients for their treatment and management.
I don't see any major changes that need to be done to the manuscript. The analysis through transcriptome and cluster analysis is robust.
Author Response
An excellent and robust manuscript on a very important subject of differentiating different sub groups of schizophrenia. I see that the authors have attempted to show that some schizophrenic patients may be struggling with chronic inflammation and how important it is to differentiate such patients for their treatment and management. I don't see any major changes that need to be done to the manuscript. The analysis through transcriptome and cluster analysis is robust.
We are pleased to hear that the reviewer considers this manuscript and its methodology to be robust and qualified for publication in its current format. We appreciate their dutifulness and honesty. No revisions have been made in response to the comments from this reviewer.